# Evolution of Anti-SARS-CoV-2 Therapeutic Antibodies

**DOI:** 10.3390/ijms23179763

**Published:** 2022-08-28

**Authors:** Juan C. Almagro, Gabriela Mellado-Sánchez, Martha Pedraza-Escalona, Sonia M. Pérez-Tapia

**Affiliations:** 1GlobalBio, Inc., 320 Concord Ave, Cambridge, MA 02138, USA; 2Unidad de Desarrollo e Investigación en Bioterapéuticos (UDIBI), Escuela Nacional de Ciencias Biológicas, Instituto Politécnico Nacional, Prolongación de Carpio y Plan de Ayala S/N, Colonia Santo Tomás, Alcaldía Miguel Hidalgo, Mexico City 11340, Mexico; 3Laboratorio Nacional para Servicios Especializados de Investigación, Desarrollo e Innovación (I+D+i) para Farmoquímicos y Biotecnológicos, LANSEIDI-FarBiotec-CONACyT, Prolongación de Carpio y Plan de Ayala S/N, Colonia Santo Tomás, Alcaldía Miguel Hidalgo, Mexico City 11340, Mexico; 4CONACyT-Unidad de Desarrollo e Investigación en Bioterapéuticos (UDIBI), Escuela Nacional de Ciencias Biológicas, Instituto Politécnico Nacional, Prolongación de Carpio y Plan de Ayala S/N, Colonia Santo Tomás, Alcaldía Miguel Hidalgo, Mexico City 11340, Mexico; 5Departamento de Inmunología, Escuela Nacional de Ciencias Biológicas, Instituto Politécnico Nacional, Prolongación de Carpio y Plan de Ayala S/N, Colonia Santo Tomás, Alcaldía Miguel Hidalgo, Mexico City 11340, Mexico

**Keywords:** COVID-19, therapeutic antibodies, variants of concern, Casirivimab, Imdevimab, Bamlanivimab, Etesevimab, Sotrovimab, Regdanvimab, Cilgavimab, Tixagevimab, Bebtelovimab

## Abstract

Since the first COVID-19 reports back in December of 2019, this viral infection caused by SARS-CoV-2 has claimed millions of lives. To control the COVID-19 pandemic, the Food and Drug Administration (FDA) and/or European Agency of Medicines (EMA) have granted Emergency Use Authorization (EUA) to nine therapeutic antibodies. Nonetheless, the natural evolution of SARS-CoV-2 has generated numerous variants of concern (VOCs) that have challenged the efficacy of the EUA antibodies. Here, we review the most relevant characteristics of these therapeutic antibodies, including timeline of approval, neutralization profile against the VOCs, selection methods of their variable regions, somatic mutations, HCDR3 and LCDR3 features, isotype, Fc modifications used in the therapeutic format, and epitope recognized on the receptor-binding domain (RBD) of SARS-CoV-2. One of the conclusions of the review is that the EUA therapeutic antibodies that still retain efficacy against new VOCs bind an epitope formed by conserved residues that seem to be evolutionarily conserved as thus, critical for the RBD:hACE-2 interaction. The information reviewed here should help to design new and more efficacious antibodies to prevent and/or treat COVID-19, as well as other infectious diseases.

## 1. Introduction

Severe acute respiratory syndrome coronavirus 2 (SARS-CoV-2) is the etiological agent of coronavirus disease 2019 (COVID-19). Since the first COVID-19 reports [1,2,3] in the province of Wuhan, China, in December 2019, SARS-CoV-2 infection rapidly spread to other countries, leading to the declaration of COVID-19 as a pandemic by the World Health Organization (WHO) on 11 March 2020 [4]. The explosive number of positive cases and the high number of fatalities during the first months of the pandemic, compounded with the devastating impact on the global economy, spurred an accelerated search for prophylactic and/or therapeutic solutions to control COVID-19. The success of antibody-based drugs, with one hundred antibodies approved by the US Food and Drug Administration (FDA) and/or European Agency of Medicines (EMA) by July 2021 to treat diverse diseases [5], including the use of antibodies to cure Ebola [6], propelled the discovery, preclinical development, and clinical testing of dozens of anti-SARS-CoV-2 antibodies [7,8]. These efforts have crystallized into the Emergency Use Authorization (EUA) by the FDA and/or EMA of nine anti-SARS-CoV-2 prophylactic and/or therapeutic antibody-based drugs (Table 1).

In parallel to the EUA of these therapeutic antibodies, hundreds of SARS-CoV-2 genetic variants have emerged as a consequence of the natural evolution of SARS-CoV-2. These variants have been routinely monitored through epidemiological research, surveillance of the genetic sequence of viruses, and laboratory studies [21,22]. Both local and international organizations have classified SARS-CoV-2 variants as Variants Being Monitored (VBM), which are those that may pose a future risk and require continued assessment; Variants of Interest (VOI), which are variants with mutations suspected to have phenotypic implications; and Variants of Concern (VOCs), which are those that have mutations with a significant increase in transmissibility, severity, and immune escape [23]. The latter have posed a formidable challenge to the use of therapeutic antibodies, in particular SARS-CoV-2 Delta (B.1.617.2), Omicron (B.1.1.529), and, more recently, Omicron sub-variants: BA.2, BA.3, BA.4, and BA.5 [24]. These genetic variants have rendered most of the FDA and/or EMA EUA antibodies ineffective to treat COVID-19 [25], resulting in a continuous race for developing new and more efficacious antibodies to control further spread of COVID-19 and, hence, prevent additional fatalities.

In this review, we first present and discuss the timeline of the VOCs emergence together with EUA of the therapeutic antibodies. A summary of the progressive lack of efficacy of the therapeutic antibodies as the VOCs emerged follows. As context for discussion of the binding and neutralizing properties of the therapeutic antibodies, we then provide an overview of the SARS-CoV-2 structure and mechanism of infection. Further, we review the sources and selection methods of the EUA therapeutic antibodies and describe the most relevant aspects of their sequences, such as IGHV and IG(L/K)V gene families, LCDR3 and HCDR3 features, somatic mutations, and epitopes. We also discuss the isotypes used in the therapeutic format and highlight the implication of Fc engineering for their mechanism of action (MoA). As SARS-CoV-2 continues to evolve, we hope this review will help to design and engineer future antibody-based drugs to treat not only COVID-19 but also other infectious diseases.

## 2. VOCs and Anti-SARS-CoV-2 Therapeutic Antibodies

Figure 1 shows the timeline of VOCs emergence and EUA of the therapeutic antibodies. Five SARS-CoV-2 VOCs have been declared by the Centers for Disease Control and Prevention (CDC), the European Centre for Disease Prevention and Control (ECDC), and the WHO. The Alpha variant (B.1.1.7 lineage) was reported in September 2020 in the UK and presented eight mutations in the S protein: H69-V70 deletions, Y144 deletion, N501Y, A570D, P681H, T716I, S982A, and D1118H [26]. Interestingly, the Beta variant (B.1.35 lineage) was reported earlier than Alpha, in May 2020 in South Africa. It has nine mutations: L18F, D80A, D215G, R246I, K417N, E484K, N501Y, A701V, and the 242–244 deletion [27]. The third VOC, Gamma variant-P.1 lineage, was first reported in November 2020 in Brazil, showing ten mutations: L18F, T20N, P26S, A138Y, R190S, K417T, E484K, N501Y, H655Y, and T1027Y [28]. The Delta variant (B.1.617.2 lineage), reported in October 2020 in India, has amino acid substitutions at T19R, G142D, L452R, T478K, P681R, and D950N, and deletions in positions 157 and 158 [29].

The latest variant, called Omicron (B.1.1.529 lineage), also known as BA.1, is by far the most divergent from the initial SARS-CoV-2 strain, being a major shift in immune evasion. Omicron was first reported in November 2021 in several cities around the world. It has five deletions in positions H69, V70, G142, V143, Y144, and N211 plus 28 mutations at positions: A67V, T95I, Y145D, L212I, G339D, S371L, S373P, S375F, K417N, N440K, G446S, S477N, T478K, E484A, Q493R, G496S, Q498R, N501Y, Y505H, T547K, D614G, H655Y, N679K, P681H, N764K, D796Y, N856K, Q954H, N969K, and L981F [30]. Omicron has recently evolved into several sub-variants, namely: BA.2, BA.3, BA.4, and BA.5 [30]. These sub-variants can be clustered together, with BA.4/5 being the most distant from the earlier BA.1 and BA.2 sub-lineages [31].

The first antibody-based drug, called REGEN-COV, received EUA in November 2020, after several VOCs emerged [10]. It was approved in an unprecedented time, e.g., less than a year after the WHO declared COVID-19 a pandemic and before any vaccine was approved. This cocktail of therapeutic antibodies is composed of Casirivimab and Imdevimab and was initially approved for indication in the treatment of adults and 12 years or older patients weighing at least 40 kg with a positive diagnosis of COVID-19 in the SARS-CoV-2 detection tests. The patients should have mild to moderate symptoms of infection and have a high risk of progression to the severe form of COVID-19 [9].

In addition to the FDA EUA of REGEN-COV as a therapeutic drug, in July 2021, the FDA authorized its use as a COVID-19 post-exposure prophylactic drug [32]. In November 2021, the EMA joined the approval of the FDA by authorizing the use of REGEN-COV as a therapeutic drug in the European Union, with the commercial name of Ronapreve [12]. This was followed by EUA of another cocktail of therapeutic antibodies. In February 2021, the FDA granted EUA to Bamlanivimab plus Etesevimab developed by Eli Lilly [33]. This company initially applied for Bamlanivimab monotherapy, but its authorization was revoked in April 2021 [34]. This therapeutic cocktail was also authorized by the FDA in September 2021 as a post-exposure prophylactic drug [35].

After the EUA of the first two cocktails of antibodies, two monotherapies reached the market. In May 2021, the FDA issued the EUA for Sotrovimab, developed by GlaxoSmithKline (GSK), for treatment of mild-to-moderate COVID-19 in adults and pediatric patients aged ≥12 years who weigh ≥40 kg with positive results of SARS-CoV-2 viral testing and who were at high risk for progression to severe COVID-19, including hospitalization or death [36]. In December 2021, the EMA recommended granting marketing authorization in the European Union to Sotrovimab [16].

The second EUA monotherapy was developed by Celltrion under the name of Regdanvimab and received EMA approval in November 2011 [12]. It should be noted that Regdanvimab has not received FDA approval yet. Regdanvimab’s indication included adult patients who do not require supplemental oxygen therapy and who are at high risk of progressing to the severe form of COVID-19 [37].

A third cocktail of therapeutic antibodies received EUA in December 2021; it was developed by AstraZeneca and is composed of Cilgavimab and Tixagevimab. Interestingly, this cocktail was approved for prophylaxis of COVID-19 in certain adults and pediatric people older than 12 years and weighing at least 40 kg. This product is only authorized for individuals not infected with SARS-CoV-2 or those not having been exposed to the virus [38].

More recently, in February 2022, the FDA issued the EUA for Bebtelovimab, also from Eli Lilly, but this time a monotherapy with indication in the treatment of moderate to mild COVID-19 in adults and patients 12 years old and older weighing at least 40 kg [39]. It is pertinent to mention that this therapeutic drug has not yet been approved by the EMA.

## 3. Efficacy of the Therapeutic Antibodies against the VOCs

The efficacy of the EUA antibodies was challenged soon after the first VOCs emerged, but the major shift in neutralization potency occurred when SARS-CoV-2 evolved into Omicron (Figure 2). Bamlanivimab plus Etesevimab lost their in vitro neutralization potency when challenged with Beta and Gamma variants, whereas Casirivimab plus Imdevimab kept their neutralizing properties and clinical activity until the emergence of Omicron. In fact, on 16 April 2021, the FDA recommended the revocation of EUA for Bamlanivimab alone due to the augmented resistance of circulant variants across the US [34]. A few months later, on 15 December 2021, the FDA no longer authorized the use of Bamlanivimab plus Etesevimab in states, territories, and jurisdictions of the US in which a combined frequency of a VOC resistance to this cocktail was reported that exceeded 5% [40]. Moreover, on 24 January 2022, the FDA limited the use of the cocktails Bamlanivimab plus Etesevimab and Casirivimab plus Imdevimab when the patients had been or were likely infected with or exposed to a susceptible variant to these treatments. This recommendation was based on the fact that both cocktails showed markedly reduced activity against Omicron [41].

Sotrovimab, on the other hand, showed a reduction in neutralization potency against Omicron BA.1 pseudoviruses of only one-fold [46] but lost 1000-fold against Omicron BA.2 sub-variant [47]. Hence, the FDA advised not to use this antibody in regions of the US where the BA.2 sub-variant predominated [48].

The neutralizing activity of Regdanvimab diminished against Beta, Gamma, and Delta. Interestingly, this antibody might still be active in humans due to the positive neutralizing results observed in animal models (ferrets and transgenic mice) [49]. However, Regdanvimab has not yet been authorized by the FDA due to a substantial reduction in its in vitro neutralization potency. Additionally, it is very likely that this antibody will have limited use.

The cocktail Cilgavimab plus Tixagevimab and monotherapy with Bebtelovimab are still recommended to be used for the treatment of COVID-19, including infection caused by Omicron sub-variants. Based on non-clinical and clinical data, the last FDA update (29 June 2022) suggested to healthcare providers the use of repeated dosing of 300 mg of Tixagevimab and 300 mg Cilgavimab every six months if the patients need protection; notice that the initial doses of these antibodies were lower (150 mg of each antibody; see below) [50].

Bebtelovimab has shown neutralization against the currently circulating SARS-CoV-2 VOCs, including Omicron BA.1 and BA.2 sub-variants. The FDA published, on 13 June 2022 [51], based on new pseudotyped virus-like particles and authentic virus data, that Bebtelovimab retains activity to Omicron subvariants BA.4 and BA.5. Therefore, Bebtelovimab seems to be the only EUA antibody broadly neutralizing SARS-CoV-2 and still efficacious against all the VOCs.

## 4. SARS-CoV-2 and Mechanism of Infection

SARS-CoV-2 is a single-stranded RNA-enveloped virus from coronaviruses (CoVs) belonging to the order Nidovirales, family Coronaviridae, and genus Coronavirus [52]. The SARS-CoV-2 RNA genome is approximately 30,000 bases long, packed in virions of 50–200 nanometers in diameter [53]. Mature virions have four structural proteins: N (nucleocapsid), E (envelope), M (membrane), and S (spike) [54]. The N protein surrounds the viral genome, protecting it from the host environment, aiding virus replication. E and M proteins regulate the intracellular trafficking and processing of the S protein [55] (Figure 3), a highly glycosylated homotrimer that covers the SARS-CoV-2 surface and is responsible for the specific recognition of human cells by the virus via human angiotensin-converting enzyme-2 (hACE-2). 

Each S monomer is made of around 1300 amino acids and is composed of two subunits: S1 and S2. S1 (residues 1–680) recognizes the hACE-2, whereas S2 (residues 681–1300) facilitates the virus fusion to the cell membrane [56]. S1 subunit contains the *N*-terminal domain (NTD; residues 13–319) and the receptor-binding domain (RBD; residues 319–541). The RBD includes the receptor binding motif (RBM; residues 437–508), which is directly involved in hACE-2-specific binding. S2 consists of the fusion peptide (FP; residues 788–806), two heptapeptide repeat sequences 1 and 2 (HR1 and HR2), a transmembrane domain (TM; residues 1213–1237), and a cytoplasm domain (CT; residues 1237–1273) [57].

RBD exhibits two conformational states. One state, called “down”, shields hACE2 binding, whereas the other state, or “up”, is accessible to hACE-2. SARS-CoV-2 entry into the cell begins when the RBD in “up” state binds to hACE-2. Between the S1 and S2 subunits, a furin-cleavage site is targeted by the furin protease in the pre-activation process. This first cleavage induces structural changes in the S2 towards a prefusion conformation. A second cleavage site, located in the FP domain, S2’, is targeted by serine proteases or cathepsins, which drives the fusion of the viral and cellular membranes, allowing the release of the SARS-CoV-2 RNA genome coated with the N protein into the cytoplasm of the targeted human cells [58].

### RBD Structure and Interaction with hACE-2

The RBD core (Figure 4) is formed by a twisted five-stranded antiparallel β sheet (β1, β2, β3, β4, and β7), connected by short loops and α-helices, and stabilized by three disulfide bonds between C336–C361, C379–C432, and C391–C525 [59]. A concave surface shaped by two small strands, β5 and β6, connected by extended loops and two α-helices, forms the RBM. Sixteen residues of the RBM (G446, Y449, Y453, L455, F456, A475, F486, N487, Y489, Q493, G496, Q498, T500, N501, G502, and Y505) and one residue (K417) outside of this region interact with the 20 residues of the N-terminal peptidase domain of hACE-2 [60]. The total buried surface area in the interaction RBD:hACE-2 is 1687 Å2, with thirteen hydrogen bonds and two salt bridges formed upon binding of these two proteins. The involvement of tyrosine residues in the RBD:hACE-2 interface is remarkable, with Y449, Y489, and Y505 forming hydrogen bonds with polar hydrogen groups from several residues of hACE-2. K417, the residue outside of RBM, is engaged in the formation of one hydrogen bond and two salt bridges [59].

The affinity of the RBD wild type or Wuhan isolate for hACE-2 has been assessed by enzyme-linked immunosorbent assay (ELISA) and surface plasmon resonance (SPR) [56]. The latter measurements, which are more accurate than ELISA [59], have been performed in diverse configurations, including covalent immobilization of the RBD or hACE-2 on the sensor chip and capture of the RBD in the context of the S1 with anti-tagged reagents. The RBD covalently immobilized to the sensor chip yielded a dissociation constant (K_D_) value of 44.2 nM [56]. When the hACE-2 was covalently immobilized to the sensor chip and RBD has flown over the chip, the K_D_ values were 20.9 nM [56] and 24.6 nM [61]. In the context of the S1, capturing the spike protein on the chip via an HIS tag yields an RBD:ACE2 K_D_ of 14.7 nM [62]. Since the K_D_ values measured with the proteins directly immobilized in the chip tend to be higher due to the lesser exposure of interacting surfaces, it is prudent to conclude that the affinity of the RBD WT for the hACE-2 is in the low nanomolar range, around 15.0 nM.

## 5. Sources of the Therapeutic Antibodies

The variable (V) regions of the therapeutic antibodies listed in Table 1 were obtained from diverse sources using different selection methods. Casirivimab was obtained from B lymphocytes isolated from convalescent patients of SARS-CoV-2 infection [63]. Its partner in the Regeneron cocktail, Imdevimab, was obtained from parallel high-throughput efforts using spleens of Regeneron Velocimmune^®^ mouse immunized with plasmid DNA expressing SARS-CoV-2 S protein and boosted with a recombinant RBD [63]. The lead antibodies were selected in such a way that recognized two distinct non-overlapping epitopes (see below).

The antibodies of the Lilly’s cocktail were obtained from two sources [64,65,66]. Bamlanivimab was obtained via high-throughput microfluidic screening of antigen-specific B cells from a hospitalized convalescent patient with COVID-19. Etesevimab was obtained by flow cytometry, gene sequencing, and genetic libraries from B lymphocytes of convalescent patients of SARS-CoV-2 infection. In the discovery campaign of the latter, 11 antibodies were generated and tested in RBD:hACE-2 blocking assays by FACS, as well as their ability to neutralize the SARS-CoV-2 in Vero E6 cells. Out of two antibodies that demonstrated potent in vitro neutralization activity, one of them, called CB6, prevented SARS-CoV-2 infection in rhesus monkeys as a prophylactic and therapeutic.

Sotrovimab was isolated from memory B lymphocytes immortalized with Epstein–Barr virus from an individual infected with SARS-CoV in 2003. The precursor of Sotrovimab, named S309, neutralized SARS-CoV-2 and SARS-CoV pseudoviruses, as well as SARS-CoV-2 virions. S309 recognized an epitope conserved within the *Sarbecovirus* subgenus (see epitope below). The scientists who isolated S309 suggested that antibody cocktails, including S309, may enhance SARS-CoV-2 neutralization and limit the emergence of immune escape mutants [67]. 

Regdanvimab is the only therapeutic antibody isolated from a phage display library [68]. The library was built with peripheral blood mononuclear cells (PBMCs) of a convalescent patient. The lead antibody, named CT-P59, was selected using RBD WT. CT-P59 neutralized SARS-CoV-2 isolates, including the first VOC reported (D614G) [68]. The therapeutic effect of CT-P59 was evaluated in three animal models, demonstrating a substantial reduction in viral titer along with relief of clinical COVID-19 symptoms [68].

Cilgavimab and Tixagevimab were obtained from B lymphocytes of two convalescent patients of SARS-CoV-2 infection from Wuhan. Close to 400 SARS-CoV-2 antibodies were isolated by flow cytometry, B memory cells enrichment, single-cell sequencing, and functional assays [69]. A subset of these antibodies bound recombinant RBD and showed neutralizing properties in a quantitative focus reduction neutralization test (qFRNT). The number of potential candidates was narrowed further down to 40 antibodies. The lead molecules, COV2-2130 and COV2-2196, precursors of Cilgavimab and Tixagevimab, respectively, were tested for their ability to block binding of the RBD to hACE-2 [70].

Finally, Bebtelovimab was isolated via high-throughput B cell screening from a COVID-19 convalescent donor. In Bebtelovimab’s discovery campaign, a total of 740,000 cells were screened by three different screening strategies, and then a machine-learning-based analysis was employed to select and rank 1,692 single antibody-secreting cells. From there, libraries of antibody genes were generated and sequenced, followed by a refined search, which led to 69 recombinant expressed antibodies. This subset of antibodies was tested in high-throughput SPR experiments to assess S protein epitope coverage, resulting in the selection of Bebtelovimab [71].

## 6. Gene Usage and LCDR3/HCDR3 Key Features

Table 2 shows the isotype, IGLV and IGHV usage, Fc modifications of the EUA antibodies. Six out of the nine therapeutic antibodies are kappa-type, whereas three, Regdanvimab, Imdevimab, and Bebtelovimab, have lambda light chains. The prevalence of kappa-type molecules, with 67% (6/9) antibodies, seems to mirror the higher proportion of functional IGKV germline genes (60%) with respect to functional IGLV germline genes (40%) in the human genome [72] rather than a bias in the selection method and/or a propensity of kappa-type antibodies to bind the RBD and/or be a specific immune response to SARS-CoV-2 infection. Moreover, one of the kappa-type antibodies is encoded by germline gene IGKV1-33, two antibodies by IGKV1-39, two antibodies by IGKV3-20, and one is encoded by IGKV4-01. IGKV1-39, 3-20, and 4-01 are highly used genes by the human immune response against diverse targets [73], and thus, these IGKV genes do not seem to be specifically selected to interact with the RBD.

Likewise, the IGHV usage appears to reflect the proportion of IGHV gene families in the human genome rather than a selection or functional bias. Antibodies encoded by the IGHV3 gene family are most prevalent in the SARS-CoV-2 therapeutic antibodies, with four out of the nine (44%) being encoded by IGHV3 germline genes. The IGHV3 gene family is the most populated in the human genome, with around 50% of the functional human IGHV genes [74,75,76,77]. Antibodies encoded by the IGHV1 family follow, with three antibodies encoded by members of this gene family, and this is the second most populated IGHV gene family in the human antibody repertoire. Interestingly, Regdanvimab and Bebtelovimab, two of the lambda-type antibodies, are paired with the only two functional IGHV genes of the IGHV2 family, suggesting a preference of anti-SARS-CoV-2 lambda-type antibodies to be paired with members of the IGHV2 gene family.

The alignment of V_H_ and V_L_ amino sequences of the nine therapeutic antibodies is shown in Figure 5. Both V domains are highly diverse in sequence and combinations of loop lengths. In the V_H_ sequences, the HCDR1 shows a predominant length of ten amino acids, with six out of nine antibodies having this length. The two remaining antibodies, Bebtelovimab and Redganvimab, have HCDR1 loops of 12 amino acids. In HCDR2, three different loop lengths are observed: 17, 18, and 20 residues. HCDR3 length ranges from nine to twenty residues, with Bebtelovimab having nine residues, and three antibodies, Casirivimab, Etesevimab, and Imdevimab, having 11 residues. The average HCDR3 length in the human antibodies is 12 amino acids [78], and, hence, four of the therapeutic antibodies can be considered as having relatively short HCDR3 loops. In the other extreme, three antibodies, Bebtelovimab, Redganvimab, and Sotrovimab, have long HCDR3s, with 18 and 20 residues.

A more detailed inspection of the HCDR3 sequences indicates that Tixagevimab has two cysteines. The X-ray structure (PDB ID: 7L7D) [79] shows that these cysteine residues make an intra-loop disulfide bond. Curiously, cysteines other than the conserved ones involved in stabilizing the V domains are considered developability liabilities [80], and the antibodies having them are deprioritized during the antibody development process [81].

V_L_ sequences are also highly diverse. Kappa-type antibodies have several loop lengths in both LCDR1 and LCDR3. Only two out of six antibodies, Casirivimab and Bamlanivimab, have LCDR3 loop lengths consistent with the predominant LCDR3 length of human kappa-type antibodies [82]. This suggests that frequent insertions and/or deletions occurred during the immune response leading to the anti-SARS-CoV-2 antibodies, which contrasts with the low number of somatic mutations. The three lambda-type antibodies, on the other hand, have three different loop lengths at LCDR3, consistent again with the high diversity of the SARS-CoV-2 therapeutic antibodies.

The number of putative somatic mutations is similar in V_H_ and V_L_, with a range from 1 to 7 mutations in V_H_ and from 0 to 6 in V_L_. The average number of mutations is 3.8 and 3.6 for V_H_ and V_L_, respectively. These mutations are mostly located in the CDRs. As a reference, the reported frequencies of mutations in the human antibody sequences are qualitatively similar for V_H_ and V_L_ [83], following an exponential distribution with a proportion of 3:2:1 mutations at the CDRs, V domains surface and V_L_:V_H_ interface, and core of the V domains, respectively. The average number of mutations per V region has been estimated for human antibodies to be around eight and five mutations for V_H_ and V_L_, respectively [83]. Therefore, the number of somatic mutations in the EUA therapeutic antibodies is below the average in humans, consistent with early observations [84] that the neutralizing antibodies, which served as substrates for engineering the therapeutic antibodies, were derived from naïve B cells.

## 7. Interaction of the Therapeutic Antibodies with the RBD

Due to the essential role of the RBD in the mechanism of infection, most of the natural anti-SARS-CoV-2 neutralizing antibodies are mounted against this region of the S1 protein [85]. In fact, there are two classifications of neutralizing antibodies based on the RBD structural recognition and their capacity to block the RBD:hACE-2 interaction. The first classification identifies four types (I–IV) [86] of neutralizing antibodies, whereas the second one identifies five types (I–V) [87]. Both classifications are similar, with the latter adding type V, which entails antibodies that recognize non-RBD SARS-CoV-2 regions, including other proteins, such as NTD and S2.

Classes I–IV define the form that groups of neutralizing antibodies interact with similar regions on the RBD surface. Class I antibodies recognize a conserved orientation of the RBM only in the “up” state conformation [88,89]. These antibodies are mostly derived from IGHV3 and IGHV1 gene families, including germline genes IGHV3-53, IGHV3-30, IGHV3-33, IGHV3-66, IGHV1-58, and IGHV1-18 [87]. Most of the therapeutic antibodies belong to this class, with four out of the nine antibodies (44%): Casirivimab, Etesevimab, Tixagevimab, and Regdanvimab (Table 3). Notice that Regdanvimab is an exception as it is encoded by a gene of the IGHV2 family.

Class II antibodies bind RBD epitopes in up and down states with no preferred orientation. These antibodies can block adjacent RBDs in the S homotrimer. The IGHV5-51, IGHV3-30, IGHV3-13, and IGHV1-69 genes are included in this class, Bamlanivimab being the only antibody classified in Class II.

Class III antibodies do not overlap with the hACE-2 binding site but sterically block the RBD:hACE-2 interaction. This interesting characteristic enables a relatively broad neutralizing profile (see below) as the antibodies in this class can recognize conserved regions around the RBM of different SARS-CoV-2 variants. The germline genes present in this class are IGHV1-18, IGHV1-2, and IGHV2-14. Imdevimab, Cilgavimab, and Bebtelovimab are Class III antibodies.

Class IV antibodies are not hACE-2 blockers but have neutralizing capacity and can bind both RBD “up” and “down” states. The germline genes involved in this class are IGHV1-18, IGHV1-2, IGHV1-46, IGHV3-3, IGHV3-23, and IGHV3-30. Sotrovimab is the only EUA therapeutic antibody belonging to this class.

Since Class V antibodies do not bind the RBD, their MoA has been proposed to be antibody-dependent cellular cytotoxicity (ADCC) and/or antibody-dependent cellular phagocytosis (ADCP), as well as interference of these antibodies with conformational changes needed for membrane fusion and/or sterically hindering with hACE-2 binding [90]. It has also been suggested [91] that, since Class V antibodies have a different MoA than types I–IV, they may be good partners of types I–IV for developing effective therapeutic cocktails [92,93,94]. Yet, no one antibody from Class V has been approved by the FDA and/or EMA as a therapeutic or prophylactic drug.

Table 3 summarizes the functional parameters of the therapeutic antibodies, including affinity, RBD:hACE-2 blockade, in vitro neutralization, and therapeutic dose. The K_D_ values range from 46 pM in Casirivimab to 6.45 nM in Etesevimab, with six of the antibodies having sub-nanomolar affinities and three, Cilgavimab, Tixagevimab, and Etesevimab, being single-digit nanomolar binders. The RBD:hACE-2 blockade activity, on the other hand, covers three orders of magnitude from 56 pM in Casirivimab to 33.6 nM in Sotrovimab, with the latter being an outlier. Moreover, the neutralization potency in all the antibodies is sub-nanomolar, ranging from 12 pM (Cilgavimab) to 970 pM (Etesevimab). The therapeutic doses range from 150 mg in Cilgavimab and Tixagevimab to 2400 mg in Regdanvimab. No correlation between these parameters, i.e., class, K_D_, blocking activity, neutralization, nor therapeutic dose, is found. Therefore, the neutralizing activity of these therapeutic antibodies in vivo seems to be an interplay of all these parameters plus the epitope and Fc modifications, discussed below.

Figure 6 shows the epitopes recognizes by the therapeutic antibodies. As a reference, we also show the mutations of the Delta and Omicron variants with respect to the RBD WT plus the binding surface of hACE-2 on the RBD. As mentioned above, two of the cocktails, Casirivimab plus Imdevimab and Bamlanivimab plus Etesevimab, lost their efficacy when challenged with SARS-CoV-2 Omicron. The third cocktail, Cilgavimab plus Tixagevimab, has diminished neutralization potency when challenged with Omicron sub-variants. Out of the three monotherapies, Regdanvimab lost neutralization potency with Omicron BA.1, whereas Sotrovimab and Bebtelovimab preserved their neutralization potency. Sotrovimab showed a significant loss in neutralization with Omicron BA.2, whereas Bebtelovimab is the only EUA antibody that has preserved its neutralization potency against all the VOCs, including the recent Omicron sub-variants BA.4/BA.5.

All nine therapeutic antibodies except Sotrovimab recognize epitopes in the RBM. Sotrovimab binds residues away from the RBD:hACE-2 interface and does not block RBD:hACE-2 interaction. This explains why Sotrovimab, despite having the least blocking activity, has a K_D_ and neutralizing potency within the range of the other antibodies. Bebtelovimab, on the other hand, mostly recognizes conserved residues in RBDs WT, Delta, and Omicron that are in the periphery of the RBD interface with hACE-2. This explains why Bebtelovimab neutralizes all the VOCs despite the large divergence among these SARS-CoV-2 variants. Binding conserved residues in the RBD as a mechanism to preserve efficacy against diverse VOCs is consistent with the analysis of a panel of 44 neutralizing antibodies of the SARS-VoC-2 WT [46], in which only six retain potent neutralizing activity against Omicron and recognize conserved residues between RBD WT and Omicron. Of note, three of these antibodies bind to the RBM, including S2K146, which triggers fusogenic conformational changes via molecular and functional mimicry of hACE-2.

These results are somewhat expected if one considers that the evolution of SARS-CoV-2 VOCs is a compromise between generating spontaneous mutations that eventually would escape the neutralizing immune response so the virus can survive and further propagate, on the one hand, and not accumulating mutations that weaken the affinity of the RBD for hACE-2 so that the virus would not be infective anymore. In fact, a study of 31,403 SARS-CoV-2 genomes randomly chosen across the world [95] identified 444 nonsynonymous mutations in RBD that yielded 49 amino acid substitutions in contact and non-contact residues with hACE-2. Comparing the location of these mutations on the structure of the RBDs of SARS-CoV-2 WT, bat-CoV, SARS-CoV, and pangolin-CoV, all of them binding human or mouse ACE-2 indicated that interactions with residues N487, Y449, G496, T500, and G502 are conserved in all the RBD mutants. Further, the authors showed that these interactions are evolutionarily conserved in *Sarbecoviruses*, which use ACE-2 as a target protein for entry into the host cells. Not surprisingly, RBD:hACE2 binding affinities and stability were maintained in all the studied mutants, consistent with the relatively conserved affinity of 5–30 nM described above for the interaction of the RBD with hACE-2, regardless of the SARS-CoV-2 variant, whether it comes from Wuhan or Omicron.

## 8. Isotypes and Fc Engineering

All EUA therapeutic antibodies are human IgG1 (hIgG1) or modifications of this isotype to enhance or attenuate the effector functions and/or half-life of the molecules (Table 2). Five out of the nine therapeutic antibodies except Etesevimab, Sotrovimab, Cilgavimab, and Tixagevimab are hIgG1 without Fc modifications. We [96] and other authors [97] have shown that IgG1 predominates the anti-SARS-CoV-2 immune response, followed by the IgG3 isotype. IgG2 and IgG4 are almost inexistent. This is a common feature of other viral infections, such as influenza, where IgG1 and IgG3 titers against the H1 are predominant in the immune response [98]. IgG1 is also the most the prevalent isotope for hepatitis C (HCV) antigens, followed by IgG3, with IgG2 and IgG4 rarely occurring or not detected [99]. Moreover, in hepatitis B infection (HBV) [100], anti-HBs neutralizing antibodies are highly biased toward IgG1 and IgG3, with only a marginal contribution by IgG2 and IgG4 isotypes. Furthermore, the long-lasting effect of IgG1 and/or IgG3 has been observed in serum samples collected 4–10 years after IFN-α therapy in patients with chronic hepatitis B [101].

The predominant role of IgG1 and IgG3 in viral infections is due to the capacity of these isotypes to elicit ADCC, ADCP, and complement-dependent cytotoxicity (CDC) [102,103]. Such immune effector functions are performed via selective Fc receptor interactions with distinct immune cell populations, such as natural killer (NK) cells, neutrophils, and macrophages, as well as the ability to bind C1q, a triggering protein of the complement pathway that leads to a cascade of events resulting in the formation of the membrane attack complex (MAC) and induction of infected cell killing. Therefore, while IgG3 is not a viable therapeutic format due to limitations in its manufacturing process, IgG1 is the isotype of choice for therapeutic antibodies when ADCC and CDC are wanted, as in the therapeutic anti-SARS-CoV-2 antibodies.

Regarding Fc modifications, Etesevimab has two mutations in the C_H_2 domain: L234A and L235A [13]. This double mutant, known as the LALA [7], reduces the effector function of the hIgG1 isotype. Conversely, Sotrovimab has double mutations M428L and N434S [15]. These alterations in the FcRn binding site, known as LS mutations [104], improve the mucosal bioavailability and increase the half-life in serum of the antibodies. Cilgavimab and Tixagevimab have mutations known as YTE and TM, which extend the half-life and reduce the FcRn interactions, respectively [105]. Importantly, neutralizing antibodies, when used as prophylactic drugs, do not require Fc effector functions. However, as mentioned above, the Fc effector functions are necessary for therapeutic applications, including reduction in the viral burden and inflammation [106,107].

## 9. Conclusions and Future Directions

Nine therapeutic antibodies have received EUA by the FDA and/or EMA to prevent and/or cure COVID-19. The initial approvals, specifically the first two cocktails of Casirivimab plus Imdevimab and Bamlanivimab plus Etesevimab, were very useful during the early stages of the pandemic when no vaccine was available to prevent SARS-CoV-2 infection. Nonetheless, the natural evolution of SARS-CoV-2 has imposed a formidable barrier to the use of EUA antibodies as VOCs have rendered most of them ineffective. Only the cocktail of Cilgavimab plus Tixagevimab has retained some of its neutralization potency when challenged with the most recently evolved VOCs, whereas Bebtelovimab is still efficacious against Omicron and Omicron sub-variants, but it is unpredictable when and where new VOCs will emerge and how Cilgavimab plus Tixagevimab and Bebtelovimab would perform against the new VOCs.

The compilation and discussion of the information presented in this review highlight some trends that might be valuable to develop a new generation of therapeutic antibodies, not only to treat COVID-19 but also for designing and optimizing therapeutic antibodies to treat other infectious diseases. First, although certain IGHV and IG(L/K)V genes seem to be overrepresented in some classes of SARS-CoV-2 neutralizing antibodies, overall, the V gene usage of the EUA therapeutic antibodies mirrors the proportion of germline genes in the human genome rather than being a product of the selection method, with a propensity to bind the RBD and/or an immune mechanism that selects certain IGV genes over others. Second, there is not a preferred HCDR3 and/or LCDR3 length as described, for instance, for anti-HIV antibodies [108], where long HCDR3 loops are a hallmark of neutralization. Third, the number of mutations is low when compared to the average in human antibodies, suggesting that the antibodies that served as a substrate to engineer the therapeutic antibodies emerged early in the immune neutralizing response. In contrast, high diversity is generated via recombination events at HCDR3 and LCDR3, leading to a significant loop lengths variation. Fourth, all the antibodies except Cilgavimab and Tixagevimab have sub-nanomolar K_D_ values, whereas the RBD:hACE-2 affinity is in the low nanomolar range (5–30 nM) for all the RBD variants, and thus, the preferred MoA is by outcompeting the RBD:hACE-2 interaction. Fifth, consistent with this MoA, the RBD:hACE-2 blocking activity and neutralization potency are also sub-nanomolar, but a clear correlation between K_D_, blocking activity, neutralization potency, and therapeutic dose is not apparent. Therefore, the MoA seems to be an interplay of affinity, epitope, and isotype. Sixth, while the neutralizing activity does not require Fc effector functions when the antibody is given as a prophylaxis, it does require Fc effector functions for therapeutic applications, pinpointing IgG1 as the preferred isotype. Seventh, antibodies that recognize residues away from the RBM, such as Sotrovimab, also lost efficacy due to the evolution of SARS-CoV-2. Eighth, antibodies that still retain efficacy, such as Bebtelovimab, bind an epitope formed by conserved residues that seem to be evolutionarily conserved and, thus, critical for the RBD:hACE-2 interaction. It is, therefore, desirable that new antibody therapeutic developments consider these residues as targets for generating future generations of therapeutic antibodies.

## Figures and Tables

**Figure 1 ijms-23-09763-f001:**
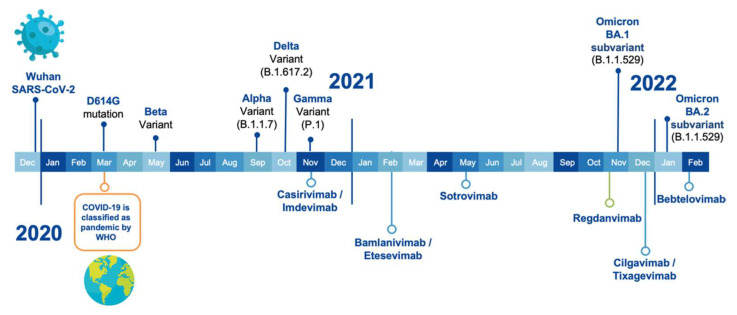
Timeline of SARS-CoV-2 VOCs emergence (on top of the time bar) along with the first EUA by the FDA and/or EMA of the therapeutic antibodies (underneath the time bar).

**Figure 2 ijms-23-09763-f002:**
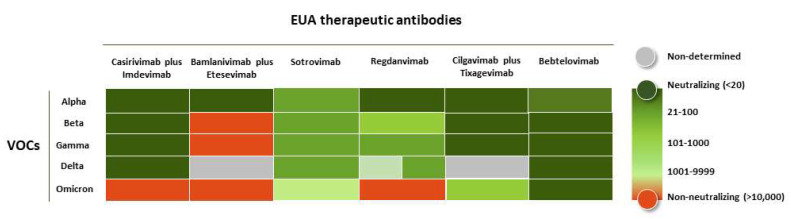
Neutralization potency of the EUA-approved anti-COVID-19 antibodies when challenged with SARS-CoV-2 VOCs. The heat map is based on the neutralization values of pseudovirus assays against SARS-CoV-2 variants [42,43,44,45]. The values of NC_50_ in ng/mL are depicted in the following color code: <20 (dark green), 21–100 (hunter green), 101–1000 (lime green), 1001–9999 (light green), >10,000 (orange).

**Figure 3 ijms-23-09763-f003:**
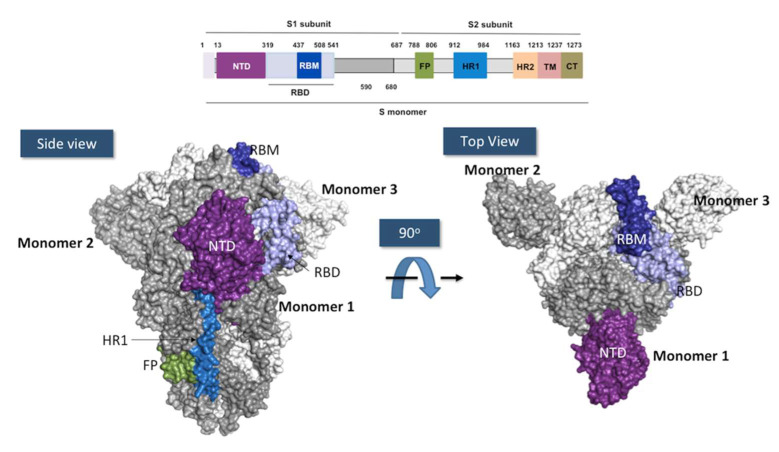
Anatomy of the SARS-CoV-2 spike protein. Above: Schematic representation of the different domains in the subunit 1 and 2 of the SARS-CoV-2 spike. NTD, N-terminal domain (in purple); RBD, receptor binding domain (in light blue); RBM, receptor binding motif (in deep blue); FP, fusion peptide (in forest green); HR1, heptapeptide repeat sequence 1 (in blue); HR2, heptad repeat sequence 2 (in melon orange); TM, transmembrane domain (in pink); CT, cytoplasmic tail (in gold). Below: Two views related by 90° rotation of the surface of Spike homotrimer protein of SARS-CoV-2 in down conformation. Each monomer is colored in white, gray and colors.. The figures were prepared using PyMOL Molecular Graphics System version 2.4.1. Schrödinger, LLC, New York (https://pymol.org/2, accessed on 16 May 2022) using the coordinates of the PDB ID: 7BNM.

**Figure 4 ijms-23-09763-f004:**
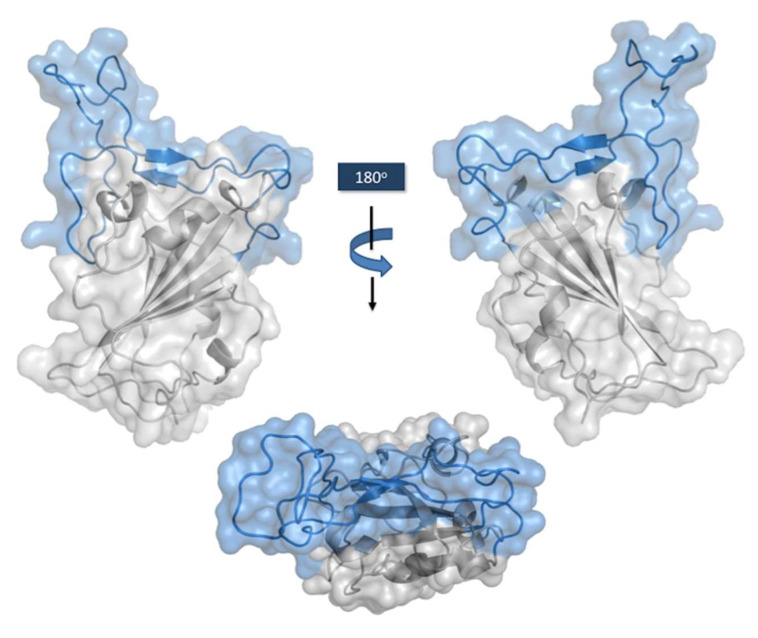
Connolly surface of the RBD with RBM motif in light blue. Side views on the top. RBM seen from the hACE-2 perspective on the bottom. The figure was generated using the coordinates with PDB ID: 7MZG in PyMOL Molecular Graphics System version 2.4.1. Schrödinger, LLC, New York (https://pymol.org/2, accessed on 16 May 2022).

**Figure 5 ijms-23-09763-f005:**
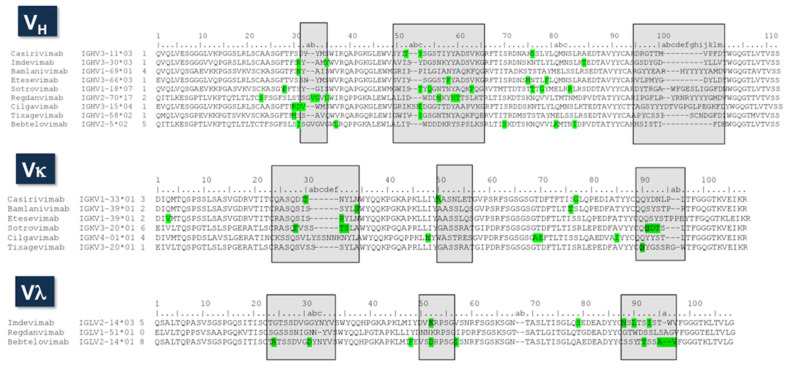
Sequence alignment of the therapeutic antibodies. The second column corresponds to the name of the IGHV, IGKV, or IGLV germline genes. The third column is the number of putative somatic mutations, indicated in green squares in the sequences. The germline genes were assigned using Ig BLAST (https://www.ncbi.nlm.nih.gov/igblast/ accessed on 16 May 2022). The CDRs are indicated in gray squares as defined using Kabat’s numbering convention.

**Figure 6 ijms-23-09763-f006:**
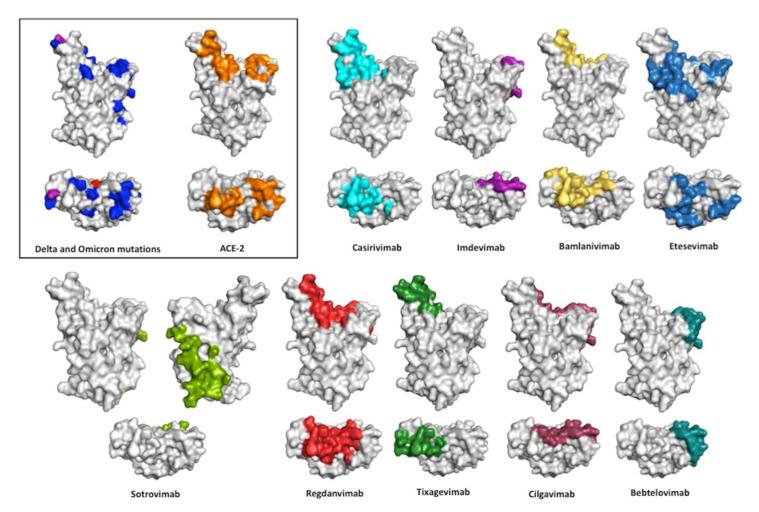
Conolly surface of SARS-CoV-2 RBD in the same view of Figure 3 showing the interface with ACE-2 and the epitopes recognized by the anti-SARS-CoV-2 therapeutic antibodies. Front and top view of the structural surface of the SARS-CoV-2 RBD shows the location of the Delta variant mutations (in red), Omicron variant mutations (in blue), and mutations in both variants (in purple). RBD residues that contact ACE-2 are indicated in orange, whereas the RBD residues protected by anti–SARS-CoV2 antibodies are indicated with different colors: Casirivimab (cyan); Imdevimab (purple); Bamlanivimab (yellow); Etesevimab (marine blue); Sotrovimab (green); Regdanvimab (red); Tixagevimab (forest green); Cilgavimab (raspberry); and Bebtelovimab (deep teal) PDB 7MZG. The figures were prepared using PyMOL Molecular Graphics System version 2.4.1. Schrödinger, LLC, New York (https://pymol.org/2, accessed on 16 May 2022).

**Table 1 ijms-23-09763-t001:** FDA and/or EMA EUA therapeutic antibodies to treat COVID-19.

INN (a)	Other Names	Commercial Name	Company	EUA	
				FDA	EMA
Casirivimab	REGN 10933	REGEN-COV, Ronapreve	Regeneron Pharmaceuticals	21 November 2020 [9,10]	11 November 2021 [11,12]
Imdevimab	REGN 10987				
Bamlanivimab	BAM, LY3819253, LY-CoV555	N/A (b)	Eli Lilly and Company	9 February 2021 [13]	EMA withdrew the application on 29 October 2021 [14]
Etesevimab	CB6, ETE, LY3832479, LY-CoV016				
Sotrovimab	S309, VIR-7831 GSK 4182136	Xevudy	GlaxoSmithKline (GSK)	26 May 2021 [15]	16 December 2021 [16]
Regdanvimab	CT-P59	Regkirona	Celltrion	N/A	11 November 2021 [12,17]
Cilgavimab	COV2-2130, AZD1061	Evusheld	Astra Zeneca	8 December 2021 [18]	24 March 2022 [19]
Tixagevimab	COV2-2196, AZD8895				
Bebtelovimab	LY-CoV1404	N/A	Eli Lilly and Company	11 February 2022 [20]	N/A

(a)—International proprietary name. (b)—N/A, non-applicable.

**Table 2 ijms-23-09763-t002:** Isotype, V regions and Fc characteristics of the FDA- and/or EMA-approved antibodies.

	Isotype	IGHV	IGHD	IGHJ	IGK/LV	IGK/LJ	HCDR3	Fc
**Casirivimab**	IgG1k	3-11 * 01	1-14 * 02	4 * 02	1-33 * 01	1 * 01	11	None
**Imdevimab**	IgG1λ	3-33 * 03	2-8 * 02	1 * 01	2-14 * 01	3 * 02	11	None
**Bamlanivimab**	IgG1k	1-69 * 09	3-16 * 01 /6 * 0104	6-01	1-39 * 01	2 * 2	16	None
**Etesevimab**	IgG1k	3-66 * 01	2-8 * 02 /4 * 01	4-01	1-39 * 01	2 * 01	11	LALA
**Sotrovimab**	IgG1k	1-18 * 01	3-16 * 01	1 * 01	3-20 * 01	1 * 01	18	LS
**Regdanvimab**	IgG1λ	2-70 * 12	1-14 * 01	6 * 02	1-51 * 01	3 * 02	18	None
**Cilgavimab**	IgG1k	3-15 * 01	3-22 * 01	4 * 02	4-1 * 02	1 * 01	20	YTE and TM
**Tixagevimab**	IgG1k	1-5 * 03	6-13 * 01	3 * 02	3-20 * 01	1 * 01	14	
**Bebtelovimab**	IgG1λ	2-5 * 02	3-3 * 02	1 * 01	2-14 * 01	3 * 02	9	None

The * indicates the allele of the genes related with the V regions of antibodies.

**Table 3 ijms-23-09763-t003:** Class and functional parameters of anti-COVID-19 therapeutic antibodies. All measurements are reported in nanomolar units.

	Class	K_D_	Blocking Assay (IC_50_)	Neutralization Assay (EC_50_)	Dose (mg)
**Casirivimab** [9,63]	I	0.046	0.056	0.04	600
**Imdevimab** [9,63]	III/IV	0.047	0.165	0.04	600
**Bamlanivimab** [13]	II	0.071	0.170	0.14	700
**Etesevimab** [13]	I	6.450	0.320	0.97	1400
**Sotrovimab** [15]	IV	0.210	33.600	0.67	500
**Regdanvimab** [37]	I	0.065	-	0.05	2400 (a)
**Cilgavimab** [18]	I	2.150	0.531	0.012	150
**Tixagevimab** [18]	III	2.180	0.318	0.06	150
**Bebtelovimab** [20]	III	0.075	0.380	0.04	175

(a) 40 mg/kg extrapolated to an average individual of 70 kg.

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
