# Peer review of "Evolution of Anti-SARS-CoV-2 Therapeutic Antibodies"

_ijms, 2022, doi:10.3390/ijms23179763_

Round 1

Reviewer 1 Report

This review covers a wide range of issues related to the use of therapeutic antibodies against SARS-CoV-2. The review is well structured and will be of interest to physicians and researchers working in the field of prevention and treatment of SARS-CoV-2. Unfortunately figures 5 and 6 are of very low quality. Except for this, the review may be published in the submitted form.

Author Response

We are grateful for the reviewer’s comments. We have improved the quality of Figures 5 and 6. Also, have done so for all the Figures. In addition, we will submit all the Figures in TIFF format as separated files to the MPDI IJMS Editorial Office in case the Figures need to be further improved during the final preparation of the manuscript for publication.

Reviewer 2 Report

This review paper illustrates the timeline of development and efficacy of neutralizing antibody agents against SARS-CoV-2.

#Conclusion

Abstract: The last part is "Here, we review the most relevant characteristics of the therapeutic antibodies including the timeline of approval, neutralization profile against the VOCs, selection methods of their variable regions, somatic mutations, HCDR3 and LCDR3 features, isotype, Fc modifications used in the therapeutic format, and epitope recognized on the receptor-binding domain (RBD) of SARS-CoV-2. The information reviewed here should help to design new and more efficacious antibodies to prevent and/or treat COVID-19 as well as other infectious diseases." The last sentence may correspond the counterpart below to stress RBD:hACE-2 interaction for the develop antibody agents.

Text: As the counterpart, the last part in 9. Conclusions and future directions "Eighth, antibodies that still retain efficacy such as Bebtelovimab bind an epitope formed by conserved residues that seems to be evolutionary conserved as thus, critical for the RBD:hACE-2 interaction. It is therefore desirable that new developments in therapeutic antibodies consider these residues as target for generating new therapeutic antibodies."

# Text chapter

As to "4.1 . RBD structure and interaction with hACE-2" please confirm the dot in between 4.1 and RBD.

#Figure legends

Figure 2. "..color code: >10,000, 1001-9999, 101-1000, 21-100, <20." needs color illustration in the legend or placing numbers in the color bar in the right.

Figure 3. "PyMOL Molecular Graphics System version 1992.4.1." needs description of its source or the reference https://pymol.org/2/

Figures 3, 4, 5, and 6. Are these original in the current manuscript? If not, reproduction permission is necessary.

Author Response

Comment:

 This review paper illustrates the timeline of development and efficacy of neutralizing antibody agents against SARS-CoV-2.

 #Conclusion

Abstract: The last part is "Here, we review the most relevant characteristics of the therapeutic antibodies including the timeline of approval, neutralization profile against the VOCs, selection methods of their variable regions, somatic mutations, HCDR3 and LCDR3 features, isotype, Fc modifications used in the therapeutic format, and epitope recognized on the receptor-binding domain (RBD) of SARS-CoV-2. The information reviewed here should help to design new and more efficacious antibodies to prevent and/or treat COVID-19 as well as other infectious diseases." The last sentence may correspond the counterpart below to stress RBD:hACE-2 interaction for the develop antibody agents.

 Text: As the counterpart, the last part in 9. Conclusions and future directions "Eighth, antibodies that still retain efficacy such as Bebtelovimab bind an epitope formed by conserved residues that seems to be evolutionary conserved as thus, critical for the RBD:hACE-2 interaction. It is therefore desirable that new developments in therapeutic antibodies consider these residues as target for generating new therapeutic antibodies."

 Response:

Added in the abstract:

“One of the most important conclusions of the review is that the EUA therapeutic antibodies that still retain efficacy against new VOCs, bind an epitope formed by conserved residues that seem to be evolutionarily conserved as thus, critical for the RBD:hACE-2 interaction.”

 Comment:

# Text chapter

As to "4.1 . RBD structure and interaction with hACE-2" please confirm the dot in between 4.1 and RBD.

Response:

Deleted this extra dot.

Comment:

#Figure legends

Figure 2. "..color code: >10,000, 1001-9999, 101-1000, 21-100, <20." needs color illustration in the legend or placing numbers in the color bar in the right.

Response:

Realized that the colors in the Figure caption got lost during the conversion to pdf format in the submission platform. We added the numbers in the color bar as suggested by the reviewer and included a color description in the caption of the Figure.  

Comment:

Figure 3. "PyMOL Molecular Graphics System version 1992.4.1." needs description of its source or the reference https://pymol.org/2/

Response:

Added in Figure 3: The PyMOL Molecular Graphics System, Version 2.4.1 Schrödinger, LLC (https://pymol.org/2/)

Comment:

 Figures 3, 4, 5, and 6. Are these original in the current manuscript? If not, reproduction permission is necessary.

Response:

We confirm that all the Figures were prepared for this manuscript and have not been published elsewhere.